# Comparative genomics of the proteostasis network in extreme acidophiles

**Katherin Izquierdo-Fiallo**[1], **Claudia Muñoz-Villagrán**[1], **Omar Orellana**[2], **Rachid Sjoberg**[1], **Gloria Levicán**[1]*

**1** Department of Biology, Faculty of Chemistry and Biology, University of Santiago of Chile (USACH), Santiago, Chile, **2** Programa de Biología Celular y Molecular, ICBM, Facultad de Medicina, Universidad de Chile, Santiago, Chile

* gloria.levican@usach.cl

**Data Availability Statement:** All data, code and protocols have been provided within the article or through supporting data files. Three supporting

## Abstract

Extreme acidophiles thrive in harsh environments characterized by acidic pH, high concentrations of dissolved metals and high osmolarity. Most of these microorganisms are chemolithoautotrophs that obtain energy from low redox potential sources, such as the oxidation of ferrous ions. Under these conditions, the mechanisms that maintain homeostasis of proteins (proteostasis), as the main organic components of the cells, are of utmost importance. Thus, the analysis of protein chaperones is critical for understanding how these organisms deal with proteostasis under such environmental conditions. In this work, using a bioinformatics approach, we performed a comparative genomic analysis of the genes encoding classical, periplasmic and stress chaperones, and the protease systems. The analysis included 35 genomes from iron- or sulfur-oxidizing autotrophic, heterotrophic, and mixotrophic acidophilic bacteria. The results showed that classical ATP-dependent chaperones, mostly folding chaperones, are widely distributed, although they are sub-represented in some groups. Acidophilic bacteria showed redundancy of genes coding for the ATP-independent holdase chaperones RidA and Hsp20. In addition, a systematically high redundancy of genes encoding periplasmic chaperones like HtrA and YidC was also detected. In the same way, the proteolytic ATPase complexes ClpPX and Lon presented redundancy and broad distribution. The presence of genes that encoded protein variants was noticeable. In addition, genes for chaperones and protease systems were clustered within the genomes, suggesting common regulation of these activities. Finally, some genes were differentially distributed between bacteria as a function of the autotrophic or heterotrophic character of their metabolism. These results suggest that acidophiles possess an abundant and flexible proteostasis network that protects proteins in organisms living in energy-limiting and extreme environmental conditions. Therefore, our results provide a means for understanding the diversity and significance of proteostasis mechanisms in extreme acidophilic bacteria.

figure and one supporting table are available with the online version of this article.

**Funding:** This study was funded by Fondo Nacional de Desarrollo Científico y Tecnológico (Fondecyt) from the government of Chile (grants 1211386 and 3200487) and Dirección de Investigación Científica y Tecnológica from University of Santiago (Dicyt-USACH), Chile. K.I.F was funded by a Conicyt Doctoral Fellowship (N°21210134). The funders had no role in study design, data collection and analysis, decision to publish, or preparation of the manuscript

**Competing interests:** The authors declare that there are no competing interests.

## Introduction

Protein homeostasis, or proteostasis, is determined by a highly flexible network of chaperones and proteases that participate in the biosynthesis, folding, trafficking and degradation of proteins [1]. This system enables the proteome to adjust to environmental changes as an adaptive response [2]. Molecular chaperones are proteins that interact with other proteins and assist with their folding and/or assembly, without forming part of their final structure [3]. Based on the mechanism of action and ATP requirements, chaperones can be classified into two groups. The first group includes the foldase chaperones, which directly participate in the folding of nascent proteins or unfolded mature proteins, using cellular ATP. Their functions determine, in many cases, that the proteins acquire their native conformation [3]. This group includes the trigger factor (TF) (the only ATP-independent foldase), DnaK, DnaJ (co-chaperone of DnaK), GroEL, GroES (co-chaperone of GroEL), HtpG, HscA, HscB, CbpA, DjlA (DnaJ homologues), MsrA and MsrB [4–6]. The second group includes holdase chaperones that act by binding to exposed hydrophobic sequences of unfolded, partially folded, or misfolded proteins, thus preventing their aggregation or degradation. While some holdase chaperones are constantly active, others are only activated under certain stress conditions [2]. Most holdases are ATP-independent and do not directly participate in restoring the native conformation of their substrates, so in many cases they depend on ATP-dependent foldase chaperones for protein refolding once the stress condition ends [2]. This group includes the chaperones ClpB, GrpE, RidA, SurA, Skp, CnoX(YbbN) and the small chaperones Hsp (heat shock protein), such as Hsp33 (HslO), Hsp20 and Hsp31 (YajL, PfpI) [7–10].

Proteins that cannot be rescued by cellular chaperones are degraded through proteolytic systems such as ClpA/P, ClpXP, ClpCP, HtrA (DegP or Do protease) and Lon (also called La protease) [11]. Proteases participate in proteostasis by degrading proteins when they are no longer required by the cell, or when they have lost their functionality. This allows the recycling of amino acids and cofactors for the synthesis of new functional biomolecules [12]. Interestingly, some proteases such as HtrA and Lon, also possess ATP-dependent foldase activity [13, 14]. This dual activity of proteases could be crucial in regulating the final fate (refolding/proteolysis) of proteins, thus determining their life cycle and recycling.

Microorganisms are constantly exposed to diverse environmental challenges such as high temperature, acidic pH, or highly oxidant conditions, which can favor the accumulation of misfolded proteins [15]. To adapt the proteome to the environmental challenges, microorganisms regulate the functionality and abundance of the determinants of proteostasis. In *Escherichia coli*, high temperature directly modulates the activity of the DnaK/J system by causing the reversible inactivation of the nucleotide exchange factor GrpE [16]. GrpE inactivation delays substrate release by slowing down the exchange of ATP/ADP by DnaK, maintaining its high-affinity state for its substrates, thus ensuring that the proteins remain bound during stress conditions. On the other hand, heat stress is usually accompanied by oxidative stress; under this dual stress, DnaK is inactivated due to the rapid and widespread decrease in cellular ATP levels [17]. In this situation, the chaperone holdase Hsp33 is activated and replaces the function of DnaK by protecting proteins, independently of ATP [18]. Once the stress ends and cellular ATP levels are restored, Hsp33 releases its substrates which are then refolded by ATP-dependent foldase chaperone systems [19]. In addition to this highly regulated mechanism, the upregulation of diverse molecular chaperones, Hsps and proteases has been described as a means of coping with temperature extremes, high osmolarity, high metal concentrations, and acidic conditions, among other factors, in a number of bacterial species [20–23].

Acidophilic chemolithoautotrophic prokaryotes that inhabit acidic environments are capable of oxidizing $Fe^{2+}$ ions and/or reduced inorganic sulfur compounds (RISC) to form $Fe^{3+}$,

and sulfate, respectively [24]. Acidophiles play an important role in bioleaching of metals from sulfide ores [25]. In addition, these microorganisms are found in natural and industrial environments where extremely harsh conditions such as very acidic pH, high osmolarity, high temperature, high load of soluble metals or severe oxidative conditions can be found [26]. All these extreme environmental conditions impose important restrictions on the growth of the microorganisms and induce stress. Although by omics studies, some knowledge concerning the nature and regulation of chaperone systems in acidophilic bacteria such as *Acidithiobacillus ferrooxidans*, and *Leptospirillum* spp. has been obtained [6, 27, 28], a detailed characterization of the components that configure the proteostasis network, and their corresponding regulation in these microorganisms, is still missing.

Whereas a few bioleaching acidophilic microorganisms have a heterotrophic or mixotrophic metabolism [29], most are strict autotrophs that must compulsorily synthesize all their cellular components by the fixation of inorganic carbon, involving a high-energy cost for the cells [30]. Furthermore, it is remarkable that most chemolithoautotrophic microorganisms satisfy all energy requirements through the oxidation of low-energy substrates like iron [31]. Based on such observations, it is predicted that these microorganisms may be subjected to a high selection pressure to optimize the synthesis and reuse of their biomolecules, especially of proteins as they are the main, and most metabolically-expensive component (approx. 55% dry weight) in the cell [32]. In this work, we performed a comparative genomic study of the proteostasis mechanisms by identifying and analyzing the variability, redundancy and distribution of the genes for the folding, repair and disaggregation of proteins, as well as proteolysis and degradation in acidophilic microorganisms. The analysis was carried out using the complete genomes of strains belonging to the phyla Pseudomonadota (Alphaproteobacteria, Betaproteobacteria, Gammaproteobacteria and Acidithiobacillia), Campylobacterota, Nitrospirota, Bacillota, Acidobacteriota, and Actinomycetota.

## Methods

### Strains and genomes

The genomes of 35 acidophilic bacteria, classified as heterotrophs (10), autotrophs (194), and mixotrophs (6), were bioinformatically inspected. The genomes of 1 acid-tolerant (*Metallibacterium scheffleri*) and 8 neutrophiles (*Gallionella capsiferriformans* ES-2, *Sideroxydans lithotrophicus* ES-1, *Mariprofundus ferrooxydans* O-1, *Sulfurimonas autotrophica* DSM16294) *E. coli* (O157:H7 and BW25113), *Pseudomonas putida* NBRC 1416, and *Salmonella enterica* serovar Thyphimurium LT2, were also included in the analysis. Genome sequences were obtained from the National Center for Biotechnology Information (NCBI) database. The strains and genome accession codes are listed in S1 Table of supporting information.

### Gene presence and gene context analysis

The genome sequences of each strain were analyzed to detect genes that code for classical cytoplasmic chaperones (Trigger Factor, DnaK, DnaJ, DjlA, CbpA GrpE, GroEL, GroES, HtpG, ClpB, HscAB, MsrAB), membrane and periplasmic chaperones (HtrA, FtsH, SurA, Skp, YidC), stress response holdase chaperones (Hsp33, Hsp20, Hsp31, SlyD, CnoX, RidA), and proteolytic systems (ClpX/A/CP, Lon) using the SnapGene 6.0.2 program. Identified genes and predicted protein products were analyzed by sequence comparison using NCBI Blast tools (https://blast.ncbi.nlm.nih.gov/Blast.cgi). The presence of conserved domains/motifs and of the ATP-binding site of each identified protein was also confirmed using the CDART (Conserved Domain Architecture Retrieval Tool) and CDD (Conserved Domains) tools from the NCBI portal (https://www.ncbi.nlm.nih.gov/cdd).

## Phylogenetic analysis

The alignment of the sequences was performed with ClustalO. The Maximum Likelihood construction algorithm was added to a Bootstrap of 1000, integrated within the MEGA11 program version 11.0.9 [33]. Tree drawing and visualization were undertaken using the software Jalview version 2.11.2.5 [34].

## Results and discussion

To gain insights into the genes and proteins that shape the proteostasis network in acidophilic bacteria, a bioinformatics search was performed as described in Methods. We carried out a genomic analysis of chaperones (foldases and holdases) and proteases using the complete genome sequences from 35 acidophilic strains belonging to 11 classes from five bacterial phyla. The identified genes of proteostasis and their corresponding products were grouped into four functional categories that included classical chaperones, AAA+ proteolytic complexes, periplasmic chaperones or proteases, and stress chaperones. Below we describe the main findings regarding these categories.

### 1. Classical chaperones

In *E. coli*, it is well described that the two main cytoplasmic ATP-dependent chaperone systems that promote refolding of protein substrates are DnaK/J/GrpE and GroEL/ES [35]. In the DnaK/J/GrpE complex, DnaK is a central foldase involved in the folding of newly synthesized polypeptides, preventing aggregation, and the refolding of misfolded and aggregated proteins [5]. DnaK binds to exposed hydrophobic regions of its substrates and promotes protein refolding in an ATP-driven process regulated by the co-chaperone DnaJ [36] or its homologues CbpA or DjlA [5]. In our analysis, the *dnaK* gene was conservatively distributed (Fig 1), with only one copy in most of the acidophiles. An exception was *Ferrovum myxofaciens* P3G and *Sulfobacillus acidophilus* TPY, which presented two non-identical copies of this gene. The gene for the co-chaperone DnaJ was present in all studied genomes, but unlike *dnaK*, it showed high redundancy in most bacterial groups, reaching a maximum of four copies in the genomes of *Sulfurimonas denitrificans* DSM1251 and *Sulfobacillus thermosulfidooxidans* DSM9293. Interestingly, genes for the DnaJ homologues CbpA and DjlA, were also found. Most of the *cbpA* copies were detected in strains of the Acidithiobacillia, and Alpha- and Gamma-Proteobacteria classes, although one copy was also identified in *Leptospirillum ferriphilum* DSM14647 and CF-1, and in *S. acidophilus* TPY. The gene that codes for DjlA was found mostly in Pseudomonadota and in *L. ferriphilum* DSM14647. It should be noted that DnaJ also presented redundancy in neutrophilic bacteria. Thus, the redundancy of the co-chaperone DnaJ and DnaJ homologues suggests the existence of variations in the activity, substrate specificity, and/or regulation of the DnaK/J/GrpE system that could potentially confer more flexibility to the proteostasis network. Also, multiple copies could contribute to suppress *dnaJ* mutations, thus representing a genetic back up to guarantee proteome maintenance in the cells under stressful conditions.

The GrpE chaperone holdase plays an important role in regulating the activity of the DnaK/J system in response to thermal stress [37]. In acidophiles, the gene encoding for GrpE was present as a single copy in all the analyzed bacterial strains (Fig 1). As in other microorganisms [38], in most acidophiles *grpE* was clustered with genes for DnaK/J/GrpE (S1 Fig). However, in the species from the Clostridia class, *grpE* clustered with genes for chaperone ClpB (not shown), which is also related to the DnaK antiaggregant system, where its function is to act on larger aggregates than those dealt with by the DnaK/J/GrpE system. Interestingly, the genomes belonging to Acidimicrobiia class (Actinomycetota phylum) harbor the genes for

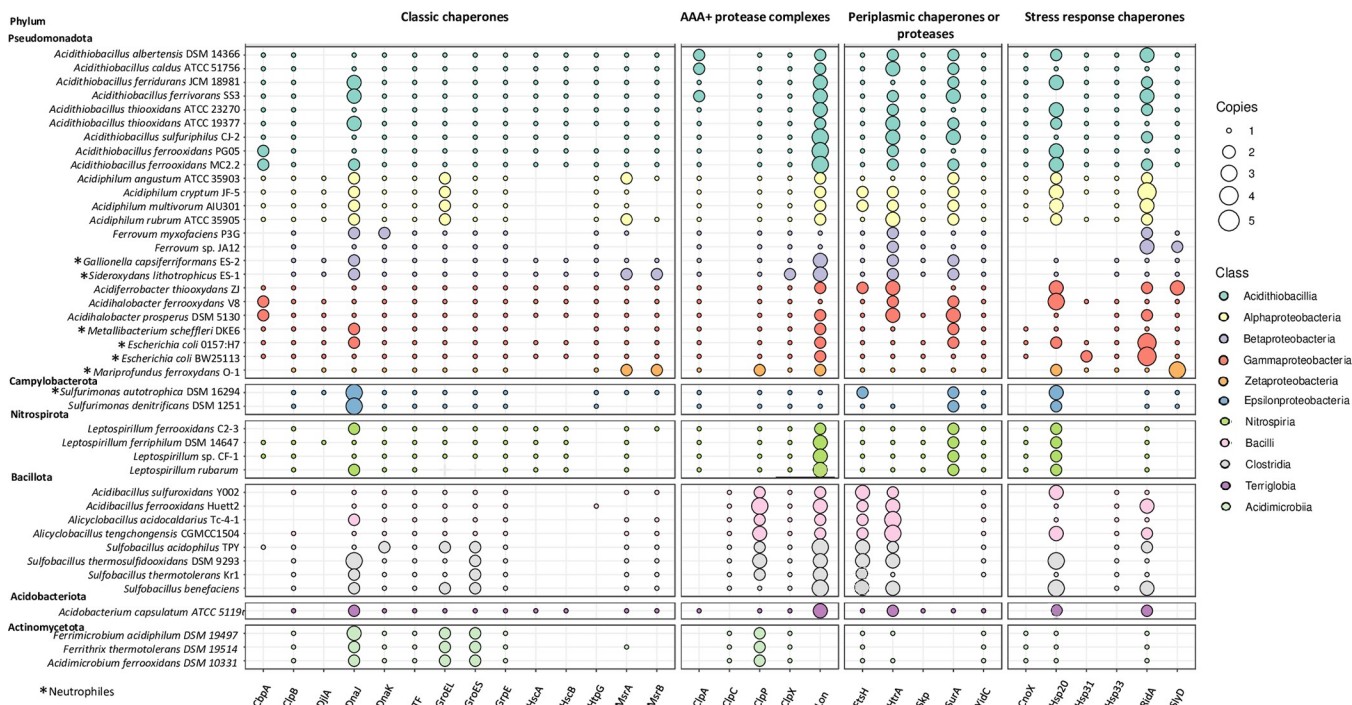

**Fig 1. Distribution and abundance of genes that code for chaperones and proteases in the genomes of acidophilic bacterial strains.** Absence of circles indicate that encoding genes were not found within the corresponding genome.

chaperones GrpE and ClpB in the same context as genes for the DnaK/J system, which could constitute an adaptive advantage that favors the coordinated expression of the antiaggregant system for protecting proteins against stress conditions. In neutrophilic model microorganisms, the genes for the DnaK/J system are found in the same context as the gene encoding the heat-inducible repressor HrcA. In *Clostridium botulinum*, the *dnaK* operon is regulated by HrcA under heat shock, NaCl and acid stress [39]. Interestingly, this repressor is absent in acidophiles belonging to classes Zeta-Proteobacteria, Epsilon-Proteobacteria from phylum Campylobacterota, and Clostridia. In addition, members of the genus *Sulfurimonas* of the Epsilon-Proteobacteria class do not have the gene for co-chaperone DnaJ in the same context as *dnaK*. These data reveal differences in the genomic organization of the DnaK/J/GrpE system, and in the gene regulatory circuits that govern the expression of the classic chaperones DnaK/J in some acidophilic bacteria.

ClpB is a high molecular weight chaperone that has an ATP-binding site that plays an essential role in the response and protection of proteins against thermal stress, as well as a great ability to rescue damaged proteins from large aggregates, together with the DnaK/J/GrpE system [40, 41]. In this study, and as shown in Fig 1, most of the microorganisms studied presented a single copy of the *clpB* gene in their genomes. Exceptions were the strains *Acidibacillus ferrooxidans* Huett2 and *Alicyclobacillus acidicaldarius* Tc-4-1, in which no copies were detected.

The GroEL and GroES chaperones form a multimeric structure that has a central compartment with ATPase activity that traps substrates and promotes folding inside [42]. GroEL/ GroES promotes the folding of almost 250 proteins in *E. coli*, representing between 10–15% of its total cytosolic proteins [43]. As shown in Fig 1, the genes for these chaperones were found in most of the genomes in a single copy; however, most representatives from Alpha-Proteobacteria, Clostridia and Acidimicrobiia classes possessed 2 copies. These data agree with previous

reports of higher redundancy and diversity of chaperones in members of the Actinomycetota phylum (formerly called Actinobacteria) [44]. Interestingly, redundancy of *groEL* and *groES* genes was detected in chemolithoheterotrophs or mixotrophs, but not in chemolithoautotrophs. The GroEL/GroES chaperone system consists of a large complex of double rings of approximately 800–1000 kDa in total [42]; thus the synthesis of multiple copies could represent a high energy cost that can only be met by cells that possess metabolisms with lower energy demands.

HtpG is a chaperone foldase that promotes the folding and activation of newly synthesized cellular proteins, and prevents aggregation and facilitates disaggregation and refolding of misfolded and aggregated proteins. The activity of HtpG increases under thermal and oxidative stress [45]. The deletion of the *htpG* gene is not lethal in bacteria and only results in higher sensitivity to the stress induced by high temperatures and reactive oxygen species (ROS) [46, 47]. The chaperone HtpG has also been shown to be influential in bacterial swarming, biofilm formation, cell division, and pathogenicity; indeed, cellular functions and mechanisms that underlie the role of the HtpG chaperone can be found in comprehensive reviews [45, 48]. Besides *E. coli*, this chaperone has been described in bacterial species such as *Clostridium tyrobutyricum* [49], *Bacillus licheniformes* [50], and *Streptomyces* spp. [51]. In the present study, the *htpG* gene was present in all proteobacterial strains as a singleton, with the exception of *A. sulfuriphilus* CJ-2 in which no copies were found. In the same way, with the exception of *A. ferrooxidans* Huett2, no genes were detected in strains from Nitrospirota, Bacillota or Actinomycetota phyla. Our results suggest that *htpG* is underrepresented in acidophiles and is restricted to representatives of Pseudomonadota.

Methionine sulfoxide reductases (Msr) are thioredoxin (Trx)-dependent oxidoreductase enzymes that repair oxidized proteins at methionine (Met-O) residues. In this way, they participate in the refolding and recovery of proteins damaged during oxidative stress [52]. MsrA and MsrB exhibit specificity for the S and R Met stereoisomers. As shown in Fig 1, in Pseudomonadota, the genes for these two proteins were present with low redundancy (1 or 2 copies). In the Alpha-Proteobacteria and Nitrospiria classes, only MsrA was found, while in the Acidimicrobiia class neither MsrA nor MsrB encoding genes were detected (only *msrA* in *Ferrithrix thermotolerans* DSM19514). Interestingly, the Beta-proteobacterium *Ferrovum myxofaciens* harbors an *msrA* gene that codes for a methionine oxidoreductase of 408 aa while other MsrA oxidoreductases possess predicted sizes ranging between 140 at 220 aa. This MsrA possesses two methionine sulfide reductase domains, at the N- and C-terminal, and a Trx domain at the N-terminal which shows similarity with the holdase CnoX from *E. coli* [35]. This non-canonical Msr from *F. myxofaciens* is similar to the fused protein MsrA/B described in *Neisseria meningitidis* which also possesses a Trx domain [53]. Interestingly, in this bacterium, the Trx domain was able to reduce both oxidized MsrA and MsrB domains, thus restoring their catalytic activity [52]. Thus, it is tempting to speculate that the atypical MsrA of *F. myxofaciens* is similar in its structure and functionality to the MsrA/B protein of *N. meningitidis*.

Finally, trigger factor (TF) is the only bacterial chaperone associated with ribosomes. TF is transiently associated with ribosomes in a 1:1 stoichiometry, binding to and acting on the most recently-nascent polypeptides emerging from the ribosome [54]. It is estimated that approximately 70% of proteins fold to their native structures after association with TF, hence a conserved presence in all bacterial genomes is expected. In agreement, in our analysis a highly conserved single copy was found in all acidophilic bacterial groups (Fig 1, S1 Table). As in other types of microorganisms, except for the strains belonging to the genus *Sulfurimonas*, in acidophiles TF is found in the same gene context along with genes for proteases ClpX and ClpP (S1 Fig). In addition, in many cases these genes were grouped with the gene that encodes the Lon protease (S1 Fig), and some genomes even presented two contiguous copies of the *lon*

gene (class Bacilli and Clostridia). In the most studied genomes belonging to the phylum Pseudomonadota, TF was also detected in the same context as a gene encoding a DNA binding beta subunit of the HU (*uh-B)* transcriptional regulator. HU plays important pleiotropic roles in DNA replication, gene regulation, translation, DNA supercoiling, and other processes, and its regulation in *E. coli* is dependent on Lon protease activity [55]. Also, it should be noted that all genomes belonging to the Acidithiobacillia and Gammaproteobacteria classes (except for *Acidiferrobacter thiooxydans* ZJ) also harbor a contiguous gene to *uh-B* that codes for HtrA (PipD) which is a peptidyl propyl cis-trans isomerase that has both proteolytic and chaperone activity [56].

In summary, classical chaperones are widely distributed in the classes from Pseudomonadota and Nitrospirota phyla. However, in Bacillota, Acidobacteriota and Actinomycetota, a sub-representation of genes for these chaperones was observed. Likely, as a compensatory adaptation, a higher redundancy of genes for GroEL and GroES systems was detected in these groups. It is interesting to note that the genes for DnaJ chaperone presented redundancy in all the studied groups, in the same way that genes for DnaJ homologues are redundant in the genomes of some groups belonging to Pseudomonadota and Nitrospirota phyla. The analysis of the gene context clearly shows that genes encoding classic chaperones and proteases are clustered, likely facilitating the regulation and coordination of the folding and degradative activities, respectively.

## 2. Proteolytic ATPase complexes associated with a variety of cellular activities (AAA+)

Here we studied the distribution and characteristics of ClpA, C, P, and X proteases, as well as the Lon protease. Clp proteins (caseinolytic proteases) are classified into class I (ClpA, ClpC, ClpD and ClpE) and class II chaperones (ClpX and ClpY) [57]. In their active form, these proteins are made up of catalytic and substrate recognition subunits. For example, the protease ClpPA consists of the catalytic ClpP and the recognition ClpA subunits. ClpP also associates with ClpX to form a protease complex with different specificity than ClpPA [58]. The structure of these proteases is like that of the GroEL/ES complex, in that they are made up of rings of several subunits that form a cavity where the target protein is degraded. In the case of ClpPA or ClpPC, the protease is made up of two central rings of seven ClpP subunits and a ring of six ClpA or ClpC subunits, respectively, at each extreme of the ClpP cylinder. These complex Clp chaperones bind to specific hydrophobic regions of misfolded or unfolded proteins to prevent their aggregation in the cytoplasm. Although under certain conditions the ClpA, ClpC and ClpX recognition subunits can also work alone as ATP-independent chaperones [59], in overall terms they constitute key regulatory components of the ATP-dependent serine protease ClpP by presenting it with the specific substrates that are destined for degradation [59]. Therefore, these systems have an important role in the death of proteins and the recycling of their components. In acidophiles, according to our results, genes for AAA+ Clp protease complexes were detected in all studied strains (Fig 1). While the *clpP* and *clpX* genes were widespread, the *clpA* and *clpC* genes were more narrowly distributed. In agreement with a previous report [57], *clpC* was detected only in Bacillota and Actinomycetota phyla.

ClpX uses multivalent strategies to discriminate between substrates that are in their native conformations or that are unfolded [58]. Therefore, when the correct folding of their substrate proteins is not achieved, a specific signal is recognized that directs them to the ClpP protease for degradation. In *E. coli*, ClpXP associates with protein aggregates [59]. As can be deduced from Fig 1, except for the neutrophile *Sideroxydans lithotrophicus* ES−1, *clpP* was present in all the species studied, and in multiple copies in strains belonging to Bacillota and

Actinomycetota, reaching a maximum of four copies in *A. ferrooxidans* Huett2 (Bacilli class). These data agree with the previously-described essential role of ClpP for representatives of the Bacillota (Firmicutes) phylum [60]. In our study, most strains containing multiple copies were classified as heterotrophs or mixotrophs (S1 Table).

Lon (La) is an ATP-dependent protease that degrades abnormal proteins or proteins that are no longer necessary for the cell [61]. In *E. coli*, this protease is responsible for 70–80% of proteolysis in the cytosol [62], and is required to maintain homeostasis and cell survival under stressful conditions. In bacteria, Lon plays a role in processes like motility, DNA replication, sporulation, and pathogenicity [63, 64]. It has a specific binding site for double-stranded DNA and proteolytically regulates the activity of some regulatory proteins like RcsA, SulA, the transcriptional activator SoxS, and UmuD [65]. In *E. coli* and *S. enterica*, among others, Lon protease activity has been reported to increase resistance to harsh conditions such as radiation, nutrient starvation, bacteriophage lysogeny, and thermal, osmotic, and oxidative stresses [63, 66, 67]. As shown in Fig 1, unlike the strains belonging to the genus *Sulfurimonas* and *Ferrovum*, and *Acidihalobacter ferrooxidans* V8, which presented a single copy, most strains contain more than two copies (2–5 copies). It should be noted that psychrotolerant strains *At. ferrooxidans* PG05 and MC2.2 [68], presented up to 4 gene copies, one of which is a truncated form that lacks the ATPase and proteolytic domains. This truncated form of Lon protease has previously been shown to be present in other microorganisms and to be active in foldase activity [69]. On the other hand, no Lon protease gene was detected in representatives of the phylum Actinomycetota. Recently, [70] showed that in the Alpha-proteobacterium *Caulobacter* sp., the lack of the Lon protease can be compensated by a variant of ClpX (named ClpX*). The complex variant ClpPX* degraded Lon substrates and suppressed defective phenotypes. Interestingly, we detected 3 copies of the *clpP* gene in actinobacterial genomes. Thus, these data suggest that in Actinomycetota, Lon activity could be potentially replaced by one or more ClpPX* variants.

In summary, the search for genes that code for proteases in acidophiles shows that they are widely distributed and that there is a redundancy of proteolytic systems that could be indicative of the importance of proteolytic degradation in the turnover of proteins. This issue could be especially relevant in organisms that possess hetero- or mixotrophic metabolism, in which protein degradation is an energetically more feasible option than in their autotrophic counterparts where, due to lower intracellular ATP levels, the degradative function could be more restrictive. The presence of variants could also represent an adaptation that confers more flexibility to the network to respond to multiple environmental challenges.

## 3. Membrane/periplasmic chaperones or proteases

In neutrophilic microorganisms, the proteostasis of the envelope proteins is carried out by periplasmic or membrane chaperones and proteases such as HtrA, SurA, Skp, YidC and FtsH. In extreme acidophiles, the pH of the periplasm lies between 2.5–3.0 [71], meaning that the proteins of this compartment, including most of the redox proteins related to iron and sulfur oxidation, carry out their functions under very acidic conditions [72]. Other stressful conditions such as the presence of $H_2O_2$ and superoxide, and high heavy metal concentrations have also been reported as having an impact on the components of the cell envelope [73, 74]. Thus, stability and tolerance to oxidants and low pH are important adaptations of proteins that are located in the periplasm and cell membrane of acidophiles. The role and nature of the chaperones that protect these proteins in these microorganisms have not yet been addressed.

The chaperone HtrA, also called protease Do, is present in all three domains of life. It is encoded in multiple copies in eubacteria, suggesting expansion by gene duplication during

evolution [75]. HtrA has a dual activity, acting as both a chaperone and a protease, so it can directly degrade or refold unfolded or misfolded proteins through the use of ATP in the periplasm. Its presence and activity are essential for bacterial survival in extreme environments [76]. This chaperone has been shown to protect the *Helicobacter pylori* proteome against multiple stresses such as heat shock, antibiotic treatment, acidic or basic stress, NaCl exposure, and elevated temperatures, preventing protein denaturation and aggregate formation [76]. Our analysis showed that, with the exception of *Ferrithrix thermotolerans* DSM19514 and the neutrophile *Sulfurimonas autotrophica* DSM16294, the chaperone HtrA is encoded in all the strains studied showing high redundancy, reaching a maximum of four copies in the Bacillota strains *Alicyclobacillus acidocaldarius* subsp. *acidocaldarius* Tc-4-1 and *Alicyclobacillus tengchongensis* CGMCC1504. Of note is that in genomes of *E. coli* and other neutrophilic bacteria, *htrA* was detected as a single copy (S1 Table). In addition, in all analyzed Bacillota, genes for chaperones SurA and Skp were not detected, suggesting that the multiple copies of *htrA* could compensate for the lack of activity of these chaperones, thus guaranteeing the protection and functionality of periplasmic proteins (Fig 1).

SurA and Skp are holdase chaperones specialized in the transport of unfolded proteins from the inner membrane to the outer membrane of Gram-negative bacteria, although it has been described that they also act as holdase chaperones in the periplasm under stress conditions, and are thus essential for cell survival [9]. SurA mediates the folding of proteins translocated to the periplasm and combines both peptidyl propyl isomerase and chaperone functions. In neutrophilic bacteria such as *E. coli*, the periplasmic chaperones HdeA/HdeB are specifically-activated by acid stress, thus preventing protein aggregation in the periplasm, while the chaperones SurA and Skp participate only in protein transport [9]. In this study, it was possible to establish that the chaperones HdeA and HdeB are not encoded in the genome of any of the strains studied, which could indicate that the chaperones SurA and Skp play a crucial role in the protection of periplasmic proteins under different types of stress. Interestingly, while *skp* was found as a singleton in most genomes of Pseudomonadota and Nitrospirota phyla, *surA* was detected in two or three copies (Fig 1). In addition, *skp* was present in a conserved context (S1 Fig) in strains from Acidithiobacillia, Beta-Proteobacteria, and *M. ferrooxydan*s, being clustered with genes for oxidoreductases which could have a role in the maintenance of the redox status of substrate proteins.

YidC is a transmembrane protein that is involved in the insertion of membrane proteins into the lipid bilayer in *E. coli* [77]. Under stressful conditions, it acts as a holdase chaperone by interacting with hydrophobic domains of unfolded or damaged proteins, so its absence results in an accumulation of aggregated or misfolded proteins in the cytoplasm and in the internal membrane. YidC can function as a ribosome receptor that directly accepts membrane proteins for their subsequent insertion [78]. Here, one copy of the YidC encoding gene was found in all the studied acidophiles (Fig 1). In most genomes, the gene was located next to those encoding the ribosomal protein L34 (*rmpH*), and others like *rnpA* that encodes the protein component of ribonuclease P involved in tRNA processing [79] (S1 Fig). Inspection of the gene context also revealed the presence of a gene that encodes YidD, a membrane protein potentially involved in the insertion of novel proteins within the membrane [80]. Therefore, as in other microorganisms, the YidC chaperone seems to have an active role in folding and insertion of membrane proteins in acidophiles.

Finally, FtsH is a highly-conserved zinc-dependent metalloprotease located in the inner membrane that belongs to the AAA+ type ATPase family. *E. coli* FtsH is the best studied of all known members and has been shown to be the only protease essential for growth and survival in bacteria. FtsH is involved in the quality control of specific membrane proteins [81], and plays an important role under heat shock when the bacterial cell has to deal with protein

aggregation [82]. In our bioinformatic analysis, at least one copy of *ftsH* was found in all genomes studied; the strains from Bacillota, *Acidiphilium* spp., and *A. thiooxydans* ZJ harbored between 2 and 3 copies (Fig 1).

Summarizing, the data obtained indicate that, with the exception of HdeA/B that were not detected, acidophiles use canonical periplasmic chaperones or proteases. However, unlike model neutrophilic bacteria such as *E. coli* or *P. putida*, which have a single copy (S1 Table), in acidophiles, genes *htrA*, *yidC* and *ftsH* were detected in multiple copies likely indicating a high level of the corresponding chaperone/protease products. This may represent a special adaptation to satisfy the requirements of folding, protection and recycling of proteins in the acid and oxidant conditions prevailing in the periplasm and external side of the membrane.

## 4. Stress response chaperones

As mentioned above, acidophiles are usually exposed to conditions like low pH, high oxidative conditions, high concentration of heavy metals, or high osmolarity, among other extreme factors. Since these challenges could induce stress, we were interested in evaluating the presence and distribution of chaperones that are generally involved in stress responses such as heat shock protein RidA, SlyD, CnoX, and Hsp.

### 4.1. Holdase chaperones

As environmental stress can lead to the accumulation of ROS and decrease the intracellular level of ATP, the activity of foldase ATP-dependent chaperones such as GroEL/ES can be affected. To cope with these stressful conditions and prevent protein aggregation, *E. coli* induces the expression of ATP-independent holdase chaperones [19, 35, 83]. These molecular chaperones are regulated at the transcriptional and/or post-translational level under stress, which allows them to respond quickly and protect the integrity of the bacterial proteome [2]. Holdase chaperones form a stable complex with damaged proteins, preventing their irreversible aggregation while the stress persists. Once the unfavorable condition subsides, the chaperones return to their inactive state and release the bound protein, which then folds itself or requires the activity of foldase chaperones. Although holdase chaperones lack refolding activity, this mechanism provides a means to prevent the accumulation of misfolded proteins and to protect the cell against the toxicity associated with protein misfolding.

RidA is a holdase chaperone that specifically responds to oxidative stress by N-chlorination, preventing protein aggregation [2]. As can be seen in Fig 1, the gene for this chaperone showed high redundancy in most of the strains, reaching up to a maximum of five copies in *Acidaldus organivorans* DX-1. Figs 2A and S2 Fig (supporting information) show the phylogenetic relationships of *ridA* gene copies and highlights the wide distribution of the gene. In addition, depending on the phylogenetic lineage, it was possible to identify two different gene contexts (Fig 2B). In Pseudomonadota, *ridA* is located next to genes *spoT* for ppGpp synthetase, and *recG* for DNA helicase RecG. The ppGpp synthetase catalyzes both the synthesis and degradation of ppGpp (guanosine 3'-diphosphate 5'-diphosphate) which is an important player in stress responses and biofilm formation [84]. The ATP-dependent DNA helicase plays a critical role in recombination and DNA repair [85]. Thus, the context of *ridA* suggests a coordinated relationship of the RidA foldase with other proteins related to protection against oxidative stress or other stressful factors, likely representing an important determinant when faced with extreme environmental conditions. Although the redundancy of *ridA* is evident in the acidophilic strains studied, this gene was also redundant in strains of neutrophilic bacteria, suggesting that its multicopy character is widely distributed, and that it could have an impact on protein abundance and/or regulation under different conditions.

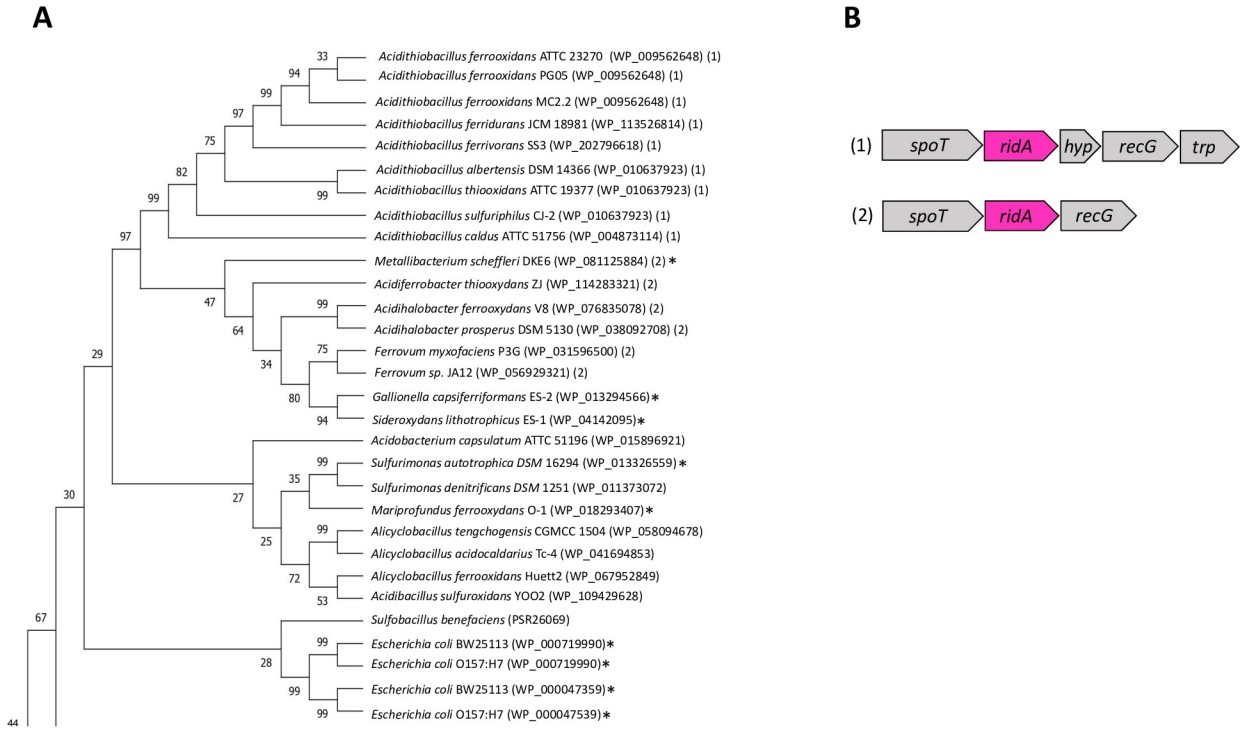

**Fig 2. Phylogenetic and gene context analysis of RidA from acidophilic bacteria. A**. Phylogenetic tree highlighting the microorganisms that share the same genetic context. **B**. Conserved *ridA* context. *spoT*: ppGpp synthetase II; *hyp*: hypothetical protein; *recG*: ATP-dependent DNA helicase; *trp*: L, D transpeptidase. Phylogenetic analysis was performed by maximum likelihood algorithm as indicated in Methods.

The chaperone SlyD is a member of the peptidyl-prolyl isomerase (PPIase) family, classified as an FK506-binding protein (FKBP) [86]. It is structurally composed of the PPIase domain that catalyzes peptidyl-prolyl cis/trans-isomerization, which accelerates the slow steps in protein folding, and the C-terminal domain with chaperone activity called IF (insert-in-flap), which prevents cytosolic protein aggregation [87]. Our results showed that *slyD* was present only in acidophilic bacteria belonging to the phylum Pseudomonadota (Fig 1). In the Acidithiobacillia class, it was present in just one copy. However, in other Pseudomonadota, genetic redundancy was detected, such as the case of *Ferrovum* sp. JA12 (2 copies), and *A. thiooxydans* ZJ (3 copies). Interestingly, *slyD* was not found in members of the Alphaproteobacteria class, all of which are classified as heterotrophs. In the genus *Acidithiobacillus*, *slyD* was neighbor to genes coding for enzymes with predicted nuclease, kinase and ferrochelatase activities, while in the microaerophilic neutrophilic iron-oxidizer *M. ferrooxydans* all four copies presented different gene contexts, two of which were close to classic chaperones MsrAB and DnaK/J/GrpE, respectively (Fig 3). Since SlyD accelerates protein folding and shortens the lifespan of protein intermediates [88], its marked gene redundancy in strict autotrophs could contribute to the efficient folding of novel, unfolded or misfolded proteins, therefore favoring a rapid and less energy-consuming response of the proteome under environmental challenges.

In oxidative stress, cysteine and methionine residues of proteins can be oxidized, which can lead to protein inactivation or misfolding. Therefore, in addition to chaperones, the oxidoreductase enzymes contribute to proteostasis by rescuing redox-sensitive residues from oxidation [20]. In this sense, CnoX (YbbN) oxidoreductase is a multidomain protein that contains a Trx domain in the N-terminal (12 kDa), and a tetratricopeptide repeat (TPR) domain in the

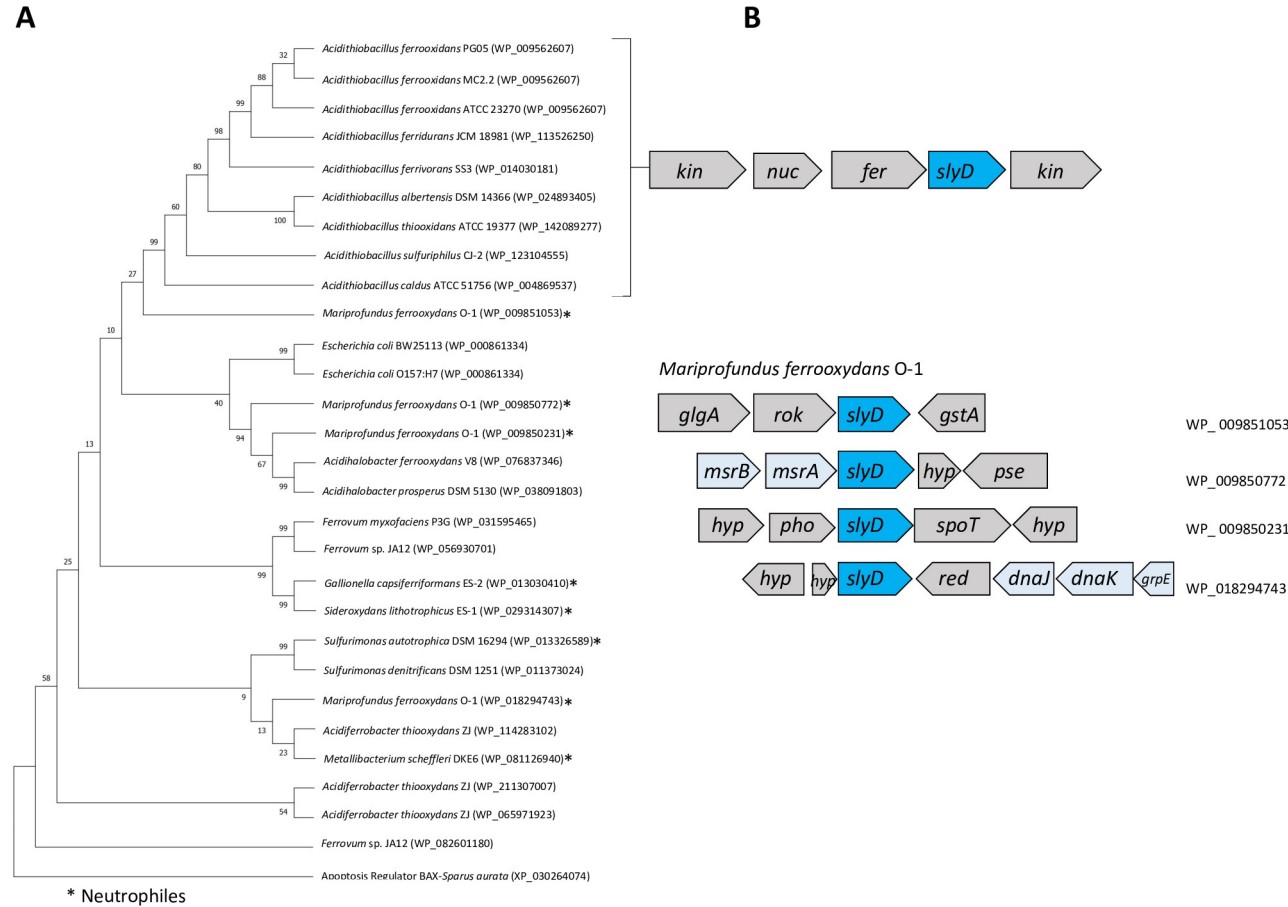

**Fig 3. Phylogenetic and gene context analysis of SlyD from acidophilic bacteria. A**. Phylogenetic tree of SlyD **B.** Genetic context of *slyD* in genomes from Acidithiobacillia class, and Zeta-proteobacterium *M. ferrooxydans* O-1. *kin*: kinase; *nuc*: nuclease; *fer*: ferroquelatase; *glgA*: glycogen synthase, *rok*: ROK protein; *gstA*: glutation transferase; *msrA*: methionine sulfoxide reductase A; *msrB*: methionine sulfoxide reductase B; *hyp*: hypothetical protein; *pse*: pseudouridine synthase; *pho*: phosphotransferase; *spoT*: ppGpp synthetase II; *red*: reductase; *dnaJ*: co-chaperone protein; *dnaK*: chaperone protein; *grpE*: nucleotide exchange factor. Phylogenetic analysis was performed by maximum likelihood algorithm as indicated in Methods.

C-terminal portion (20 kDa). The Trx domain consists of four stranded beta-sheets flanked by three alpha-helices, and the redox-active CXXC motif, which is involved in the reduction of oxidized thiols in proteins, whilst the TPR domain consists of 3 to 16 repeated regions that provide a concave groove for protein-protein interactions and protein folding processes. It has been reported that CnoX-deficient strains are sensitive to heat stress [89]. More recently, CnoX was described as a key protein in the bacterial response to hypochlorous acid (HOCl) in *E. coli* [35]. Therefore, CnoX has a dual function that prevents irreversible protein aggregation and protects cellular proteins from hyperoxidation. After stress, CnoX transfers its substrates to DnaK/J/E and GroEL/ES for refolding, so CnoX is the only known holdase to date that directly cooperates with the essential GroEL/ES machinery. In this study, our results showed that all strains studied belonging to the Acidithiobacillia and Alpha-Proteobacteria classes, and Nitrospirota and Actinomycetota phyla had a single copy of *cnoX* (Fig 1; Fig 4A). In Pseudomonadota, *cnoX* was detected in the same gene context as the gene for the protease HtrA (Fig 4). As previously reported [90], in representatives of the Nitrospirota class, a gene for a shorter CnoX-like protein that conserves the Trx fold domain was found adjacent to genes coding for GroES/EL and a protease.

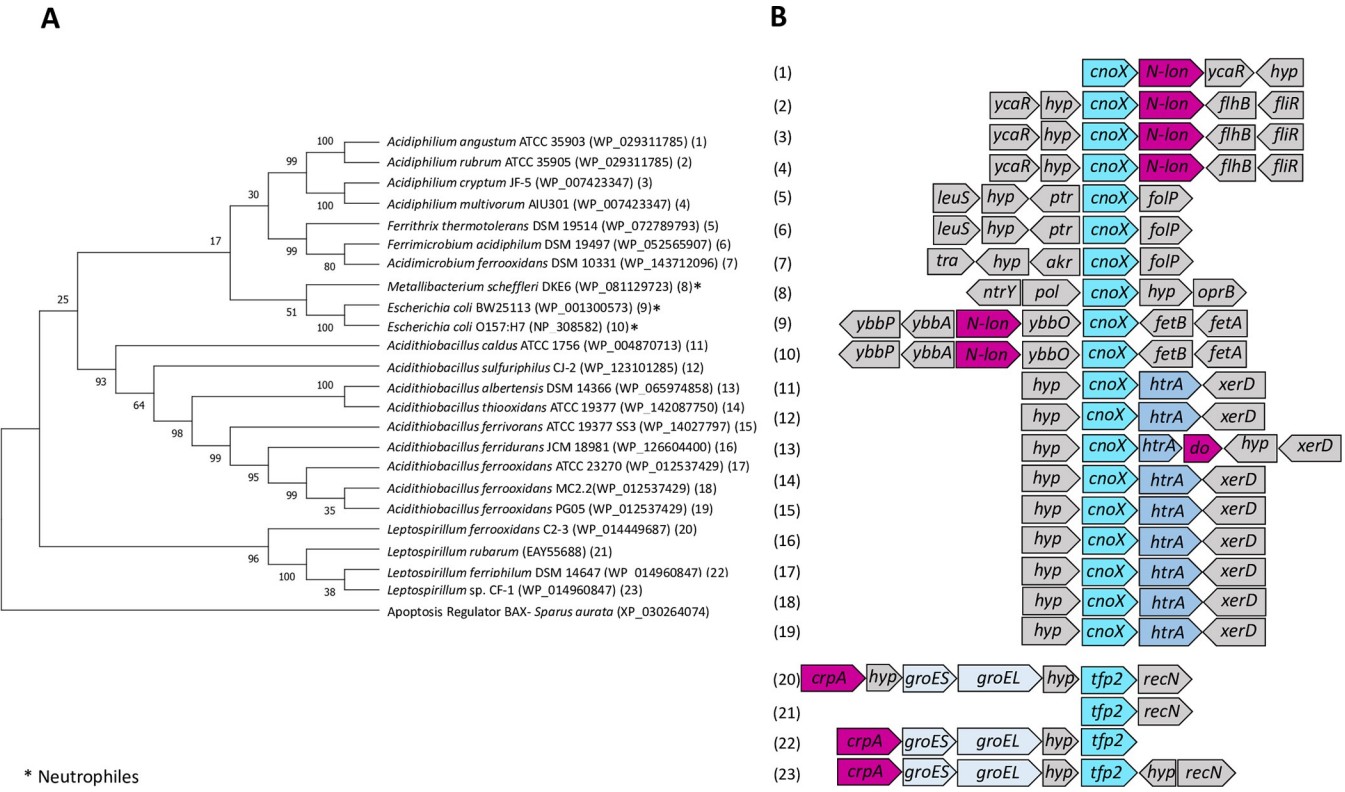

**Fig 4. Phylogenetic and gene context analysis of CnoX from acidophilic bacteria. A**. Phylogenetic tree of CnoX **B**. Gene context of *cnoX* and its homologue *tfp2*. N-*lon*: N-terminal Lon protease; *hyp*: hypothetical protein; *ycaR*: protein YcaR; *flhB*: flagellar biosynthesis protein; *fliR*: flagellar biosynthesis protein; *leuS*: leucine tRNA ligase; *ptr*: phosphotransferase; *folP*: dihydropteroate synthase; *tra*: transferase; *akr*: aldo-keto reductase; *ntrY*: nitrogen regulation protein; *pol*: DNA polymerase; *oprB*: carbohydrate porin; *ybbP*: uncharacterized ABC transporter permease; *ybbA*: putative ABC transporter ATP-binding protein; *ybbO*: uncharacterized oxidoreductase; *fetB*: probable iron export permease; *fetA*: putative iron ABC exporter ATP-binding; *htrA*: serine protease; *xerD*: site-specific tyrosine recombinase; *do*: protease; *crpA*: protease; *groES*: chaperone GroES; *groEL*: chaperone GroEL; *tfp2*: thioredoxin-fold protein 2; *recN*: DNA repair protein. Phylogenetic analysis was performed by maximum likelihood algorithm as indicated in Methods.

## 4.2. Small heat shock protein (sHsp)

The sHsps are low-molecular-weight holdase chaperones, ranging from 12 to 43 kDa that were initially described as heat shock proteins [91]. However, it is currently known that they can have a protective role against a variety of stressful environmental conditions in both eukaryotic and prokaryotic cells [91, 92]. Some examples of the most studied Hsps in prokaryotes are Hsp20, Hsp31, Hsp33, and Spy [93–96].

The chaperones Hsp31 and Hsp33 play a major role in the oxidative stress response in *E. coli*, preventing protein aggregation [2]. Hsp31 is also involved against acid stress [10]. As shown in Fig 1 and Fig 5A, the genes of both chaperones were present as single copies in the genome of most Pseudomonadota. The Hsp33 encoding gene was also detected in Bacillota. In members of the genus *Acidithiobacillus*, and *A. ferrooxydans* V8, the gene for Hsp31 was found in the same context as genes for the ClpA protease and the Clp protease adaptor ClpS [57] (Fig 5B).

The Hsp20 chaperone is constitutively active and forms stable complexes with its substrates under stress conditions [97]. *In vitro* experiments have shown the ability of this protein to suppress protein aggregation at elevated temperatures in *Deinococcus radiodurans* [97]. In *E. coli*, Hsp20 suppressed inactivation of several enzymes by heat, ROS, and freeze-thaw [98, 99]. In bacteria, the gene that encodes Hsp20 is under the control of RpoH or RpoS, which are master regulators of heat shock and general stress responses, respectively [99, 100]. In *Azotobacter*

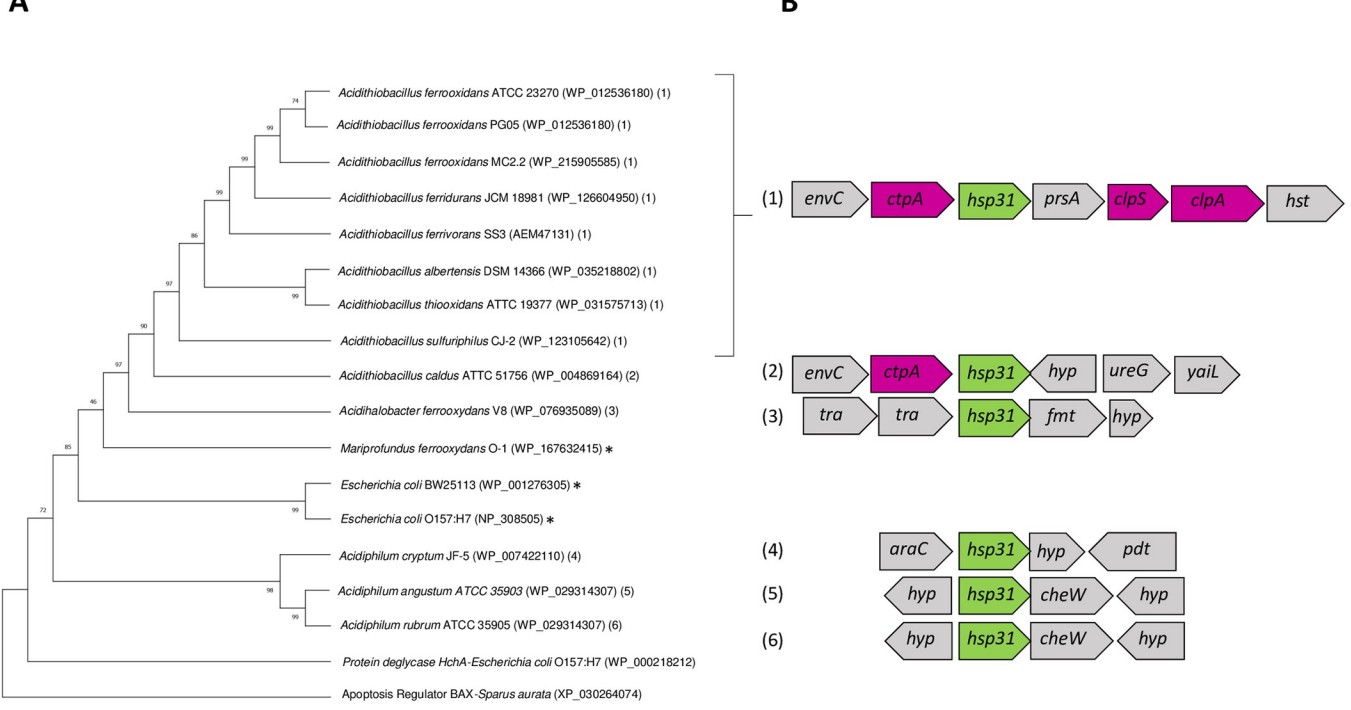

**Fig 5. Phylogenetic and gene context analysis of Hsp31 from acidophilic bacteria. A**. Phylogenetic tree of Hsp31 **B.** Gene context of *hsp31* in acidophilic microorganisms. *hst*: homoserine transferase; *tra*: transferase; *fmt*: methionyl-tRNA formyltransferase; *pdt*: phosphodiesterase; *envC*: murein hydrolase activator; *ctpA*: copper exporting P-type ATPase; *prcA*: proteosome subunit alpha; *clpS*: ATP-dependent Clp protease adapter ClpS; *clpA*: ATP-dependent Clp protease; *hst*: homoserine transferase; *hyp*: hypothetical protein; *ureG*: urease accessory protein UreG; *yaiL*: DUF2058 domain containing protein; *tra*: transferase; *yagQ*: molybdenum cofactor insertion chaperone YagQ; *panE*: 2-dehydropantoate 2-reductase; *thiL*: thiamine-phosphate kinase; *araC*: L-arabinose operon transcriptional regulator; *cheW*: chemotaxis protein. Phylogenetic analysis was performed by maximum likelihood algorithm as indicated in Methods.

*vinelandii*, Hsp20 is also involved in desiccation resistance [100]. In our analysis, Hsp20-encoding genes are widely-distributed and highly-redundant. The gene was present in most acidophiles with an average of 2 copies per genome, reaching a maximum of 4 copies per genome. Exceptions that did not contain *hsp20* were *Acidithiobacillus ferrivorans* SS3, *Ferrovum* spp., and *S. acidophilus* TPY. In members of the *Acidithiobacillus* genus, three non-identical *hsp20* gene copies were detected (*hsp20.1*, *hsp20.2*, and *hsp20.3*). The *hsp20.1* and *hsp20.3* genes were also identified in representative genomes of the Nitrospiria class. These copies clustered into three discrete groups (named I, II and III) (Figs 6A and S3 Fig). Additionally, in different genomes, each copy presented a highly conserved and particular gene context. The *hsp20*.1 was adjacent to a gene for an alcohol dehydrogenase with a GroES domain (Fig 6B), *hsp20.2* was close to genes encoding HtrA or YidC proteases, and *hsp20.3* was next to a gene coding for a Lon protease. The high redundancy of *hsp*20 suggests that this holdase chaperone plays a pivotal role in protein protection and cell proteostasis in acidophiles. The proximity of *hsp20* to protease encoding genes suggests a coordinated regulation of gene expression, and thus likely of the activities of the corresponding proteins.

## Conclusions

The genomic study of the proteostasis machinery involving folding, repair, disaggregation, proteolysis and degradation of proteins revealed that acidophiles use widely distributed

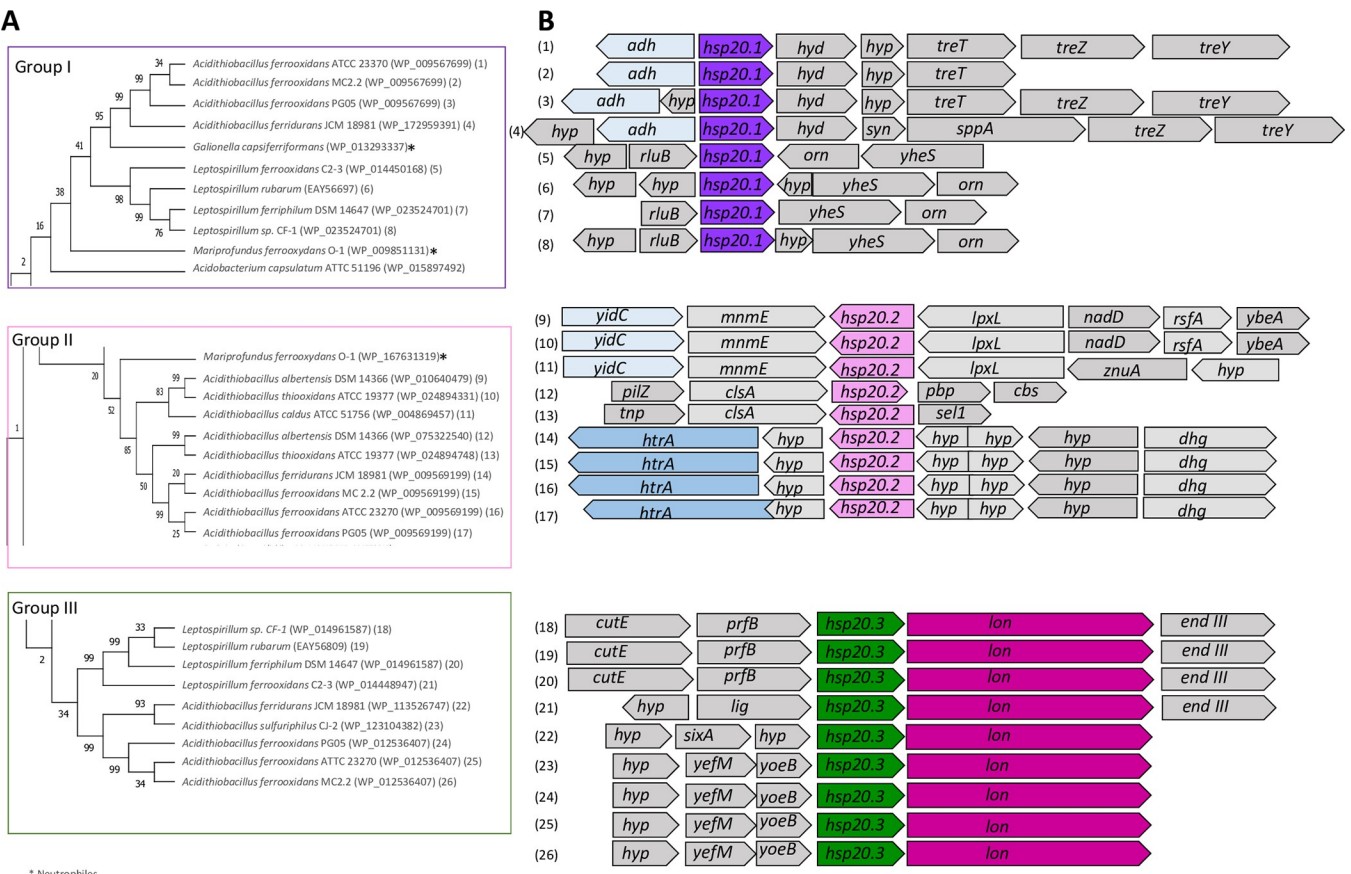

**Fig 6. Phylogenetic and gene context analysis of Hsp20 from acidophilic bacteria. A**. Phylogenetic tree of Hsp20 highlighting groups I, II and III. **B.** Gene context of *hsp20* from group I (*hsp20.1*), group II (*hsp20.2*) and group III (*hsp20.3*); *adh*: alcohol dehydrogenase; *hyd*: hydrolase; *hyp*: hypothetical protein *treT*: trehalose synthase*; treZ*: malto-oligosyltrehalose trehalohydrolase; *treY*: maltooligosyl trehalose synthase; *syn*: synthase; *sppA*: protease IV; *luxR*: transcription factor LuxR; *csp*: cold shock protein; *kin*: kinase; *ABC tra*: ABC transporter; *rluB*: ribosomal large subunit pseudouridine synthase B; *orn*: oligoribonuclease; *yheS*: putative ATP-protein YheS*; yidC*: membrane protein insertase YidC*; mnmE*: tRNA modification GtPhase MnmE*; lpxL*: lipid A biosynthesis lauroyl transferase; *nadD*: nicotinate-nucleotide adenyltransferase*; rsfA*: ribosomal silencing factor RsfS*; ybeA*: ribosomal RNA large subunit methyltransferase H*; pilZ*: flagellar brake protein YcgR; *clsA*: cardiolipin synthase A; *pbp*: penicillin binding protein*; cbs*: putative signal-transduction protein with CBSd; *tnp*: transposase; *sel1*: Sel1 repeat family protein; *htrA*: serine protease HtrA; *dhg*: dehydrogenase; *cutE*: apolipoprotein N-acyltransferase; *prfB*: peptide chain release factor *2; lon*: Lon protease; *end* III: endonuclease III; *lig*: ligase; *sixA*: phospohistidine phosphatase SixA*; yefM*: antitoxin YefM*; yoeB*: toxin YoeB. Phylogenetic analysis was performed by maximum likelihood algorithm as indicated in Methods.

canonical systems (Fig 7). The encoding genes were found to be broadly dispersed in Pseudomonadota, Nitrospirota, and Acidobacteriota. A remarkable feature of acidophilic bacteria was the redundancy of genes for certain systems including, for example, the protease Lon and the chaperone DnaJ. Also, the gene encoding the periplasmic chaperone/protease HtrA and chaperone SurA showed a redundancy which is likely necessary to confer protection to proteins and to ensure the degradation of mis/un-folded proteins in the harsh conditions that persist in the periplasm. Furthermore, the redundancy of genes for holdase chaperones Hsp20 and RidA may represent an adaptive strategy to protect proteins and to avoid their aggregation under low-ATP conditions. Bacillota and Actinomycetota also showed additional gene redundancy for the protease ClpP, the chaperone GroES, GroEL, and the holdase FtsH while other systems like protease ClpA, chaperones CbpA, DjlA and HtpG, and holdases Skp, SurA, Hsp31 and SlyD were absent. It is also noticeable that genes for holdase chaperones in acidophiles were clustered with genes encoding proteases suggesting a coordinated regulation of activities

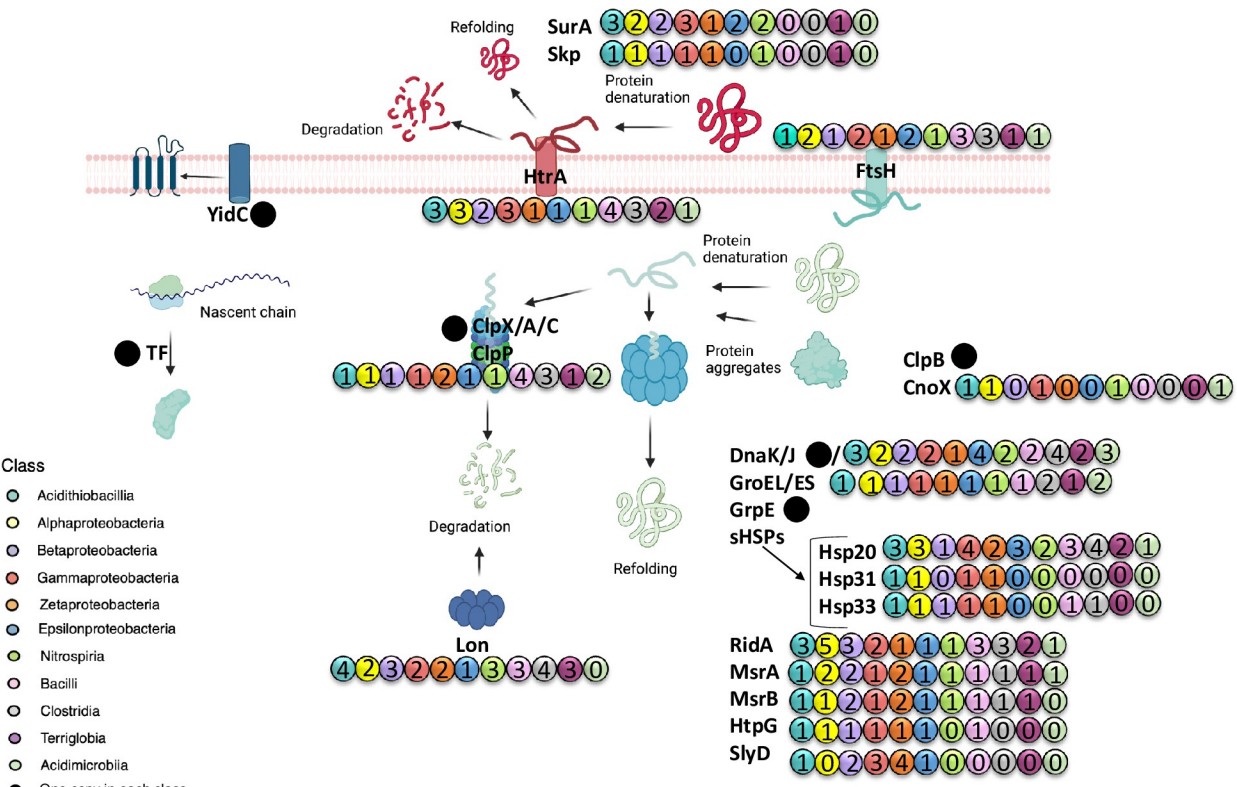

**Fig 7. Schematic representation of predicted proteostasis networks in acidophiles.** The prediction of genetic products was derived from 40 genomic sequences of heterotrophic, autotrophic, and mixotrophic acidophilic microorganisms. Numbers represent the copies detected in every class. Colors represent the corresponding bacterial class.

related to protein protection and degradation, thus avoiding accumulation and toxicity of unfolded proteins. The copies of *hsp20* and *ridA* genes were non-identical, and present in different genetic contexts suggesting their functional differentiation that could significantly contribute to providing a higher flexibility of responses to various environmental challenges. In this way, in our study the holdases Hsp20 and RidA arose as important candidates of tolerance in acidophilic microorganisms. In addition, the presence of atypical protein variants of members of this system, such as those detected for MsrA and ClpPX could confer additional capabilities to the proteostasis network in those microorganisms that contain them. The findings of this study provide the first clue about the diversity and abundance of genes related to the proteostasis network in acidophilic bacteria and raise questions and perspectives about the functionality and relevance of these elements in order to thrive in extreme acidic environments.

## Supporting information

**S1 Fig. Conserved gene contexts of different chaperones and proteases.**
(TIF)

**S2 Fig. Phylogenetic tree of RidA in acidophiles by maximum likelihood method.**
(PDF)

**S3 Fig. Phylogenetic tree of Hsp20 in acidophiles by maximum likelihood method.**
(PDF)

**S1 Table. Accession numbers of predicted proteins involved in the proteostasis network in acidophiles.**
(XLSX)

## Author Contributions

**Conceptualization:** Claudia Muñoz-Villagrán, Omar Orellana, Gloria Levicán.

**Formal analysis:** Katherin Izquierdo-Fiallo, Claudia Muñoz-Villagrán, Rachid Sjoberg.

**Investigation:** Katherin Izquierdo-Fiallo, Claudia Muñoz-Villagrán.

**Methodology:** Katherin Izquierdo-Fiallo, Claudia Muñoz-Villagrán, Gloria Levicán.

**Writing – original draft:** Katherin Izquierdo-Fiallo, Claudia Muñoz-Villagrán.

**Writing – review & editing:** Omar Orellana, Gloria Levicán.

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
