## [Decision Letter · Decision Letter 0]

25 Jun 2023

PONE-D-23-16802Comparative genomics of the proteostasis network in extreme acidophilesPLOS ONE

Dear Dr. Levicán,

Thank you for submitting your manuscript to PLOS ONE. After careful consideration, we feel that it has merit but does not fully meet PLOS ONE’s publication criteria as it currently stands. Therefore, we invite you to submit a revised version of the manuscript that addresses the points raised during the review process.

We look forward to receiving your revised manuscript.

Kind regards,

RAJA AADIL HUSSAIN BHAT

Academic Editor

PLOS ONE

Journal Requirements:

"This study was funded by Fondo Nacional de Desarrollo Científico y Tecnológico (Fondecyt) from the government of Chile (grants 1211386 and 3200487) and Dicyt-USACH from University of Santiago, Chile. K.I.F was funded by a Conicyt Doctoral Fellowship (N°21210134)."

3. Please expand the acronym “Dicyt” (as indicated in your financial disclosure) so that it states the name of your funders in full.

"The authors declare that there are no competing interests."

Additional Editor Comments:

Dear Dr. Gloria Levicán

Thank you for submitting your manuscript to Plos One.

I have completed my evaluation of your manuscript. The reviewers recommend reconsideration of your manuscript following major revision. I invite you to resubmit your manuscript after addressing the comments below. Moreover, you are advised to Improve the DPI of images.

When revising your manuscript, please consider all issues mentioned in the reviewers comments carefully: please outline every change made in response to their comments and provide suitable rebuttals for any comments not addressed. Please note that your revised submission may need to be re-reviewed.

Reviewers' comments:

Reviewer's Responses to Questions

**Comments to the Author**

1. Is the manuscript technically sound, and do the data support the conclusions?

Reviewer #1: Partly

Reviewer #2: Yes

2. Has the statistical analysis been performed appropriately and rigorously? 

Reviewer #1: N/A

Reviewer #2: Yes

3. Have the authors made all data underlying the findings in their manuscript fully available?

Reviewer #1: Yes

Reviewer #2: Yes

4. Is the manuscript presented in an intelligible fashion and written in standard English?

Reviewer #1: Yes

Reviewer #2: Yes

5. Review Comments to the Author

Reviewer #1: In this research, a comparative genomic analysis of the genes encoding classical, periplasmic and stress chaperones, and the protease systems was carried out. The research provides valuable information about the diversity and significance of proteostasis mechanisms in bacteria.

Major issues:

Many of the analyzed genomes do not belong even to moderately acidophilic microorganisms, while the authors state that these bacteria are extreme acidophiles. There are also incorrect statements about some autotrophic, mixotrophic, or heterotrophic bacteria. For instance, Mariprofundus ferroxydans is an obligate chemolithoautotroph that oxidizes reduced Fe from a variety of substrates at pH 5.5–7.2 (Singer et al., 2011) https://journals.plos.org/plosone/article?id=10.1371/journal.pone.0025386 . Therefore, it is not an acidophile but a neutrophile and not a mixotroph but an autotroph.

Gallionella capsiferriformans is not an acidophile (grows at circumneutral pH) (Fabisch et al., 2015) https://onlinelibrary.wiley.com/doi/10.1111/gbi.12162 .

Metallibacterium scheffleri is acid-tolerant (optimum pH for growth is 5.5), it inhabits acidic and some neutral environments. It grows with casein as the only carbon and energy source. The strain DKE6(T) was not able to oxidize iron or thiosulfate. Iron reduction was detected. Therefore, it is not obligately autotrophic. https://www.microbiologyresearch.org/content/journal/ijsem/10.1099/ijs.0.042986-0#tab2

https://academic.oup.com/femsec/article/93/3/fix011/2962728 .

Sulfurimonas autotrophica is neutrophilic; it grows at 4·5–9·0 (optimum pH 6·5) https://www.microbiologyresearch.org/content/journal/ijsem/10.1099/ijs.0.02682-0

Alicyclobacillus acidocaldarius subsp. acidocaldarius Tc-4-1 and Alicyclobacillus tengchongensis are heterotrophic microorganisms, not autotrophic.

Acidimicrobium ferrooxidans oxidizes ferrous iron and, therefore, should not be characterized as heterotroph.

Please check relevant information about all bacteria used in this analysis and revise the text throughout the manuscript, including l. 274–275 and Table S1. If this research is focused on extreme acidophiles, only bacterial genomes of acidophilic microorganisms should be included. Otherwise, the concept and focus of this research should be changed.

Reviewer #2: It would be beneficial to begin the introduction with references to the influence of acidic pH, high concentrations of dissolved metals and high osmolarity on proteostasis Network in order to enhance the value of the study. This article can be included after some revisions.

1. The resolution of most Figures are too low to print for publisher.

2. Why did authors choose the 40 acidophilic bacteria mentioned in the manuscript as the research object, and what are the unique features of these bacteria.

3. The content is plentiful, but some part of the reference literatures is kind of obsolete (in 5 years). Key publications should be cited as completed as possible.

6. PLOS authors have the option to publish the peer review history of their article (what does this mean?). If published, this will include your full peer review and any attached files.

Reviewer #1: No

Reviewer #2: No

---

## [Author Response · Author response to Decision Letter 0]

15 Aug 2023

Answer to the Reviewers PONE-D-23-16802

We thank the effort and work of the reviewers in order to improve the quality and clarity of our manuscript. We have followed the advices and below you can find answers to the point raised by the reviewers 1 and 2. It is of note that Figures 2 and 6 have been modified so that only a part of the phylogenetic tree is presented, while the full tree is attached in supporting information as figures S2 and S3.

Reviewer #1

Many of the analyzed genomes do not belong even to moderately acidophilic microorganisms, while the authors state that these bacteria are extreme acidophiles. There are also incorrect statements about some autotrophic, mixotrophic, or heterotrophic bacteria.

Please check relevant information about all bacteria used in this analysis and revise the text throughout the manuscript, including l. 274–275 and Table S1. If this research is focused on extreme acidophiles, only bacterial genomes of acidophilic microorganisms should be included. Otherwise, the concept and focus of this research should be changed.

Answer. We have carefully inspected the information related to all microorganisms analysed. Then, when corresponded we modified the categories as suggested by reviewer 1. The information related to metabolism and acidophilic/neutrophilic character of each bacterium was modified in the S1 Table and also in the text of the whole manuscript. One exception was Acidimicrobium ferrooxidans which oxidizes ferrous iron, but also fix CO2 and organic carbon as carbon sources, for that it was classified as mixotrophic (not heterotroph as suggested by the reviewer 1). Below, we have attached a list that indicates the character of each microorganism and contains the corresponding reference. 

The revised version of the manuscript clearly indicates in the text the acidophilic or neutrophilic character of the bacteria. As indicated the main purpose is characterizing the proteostasis network in the acidophilic microorganisms; thus we have carried out the analysis using acidophilic bacteria genomes. However, the neutrophilic bacteria were also used for comparison purposes. The general conclusion that was initially stated was not affected or contradicted.

Reviewer #2: 

a. It would be beneficial to begin the introduction with references to the influence of acidic pH, high concentrations of dissolved metals and high osmolarity on proteostasis Network in order to enhance the value of the study. This article can be included after some revisions.

Answer. The referred information by the reviewer 2 was in fact included in the third paragraph of the original manuscript. Now, we have slightly modified the text to include osmolarity as an influential factor on the proteostasis. However, we prefer to keep the original order of the Introduction section, where a general description about proteostasis concept was given first, and then a brief discussion about the effect of environmental factors. We, hope the reviewer can accept to keep this structure. 

b. The resolution of most Figures are too low to print for publisher.

Answer. We have improved the quality of the figures. We hope it meet the requirements of the journal to this regard. In additional, the Figures 2 and 6 were have been modified so that only a part of the phylogenetic tree is presented, while the full tree is attached in supporting information as figures S2 and S3.

c. Why did authors choose the 40 acidophilic bacteria mentioned in the manuscript as the research object, and what are the unique features of these bacteria.

Answer. The number of total acidophilic bacteria described so far amounts to a few tens; so the number itself is quite small. In addition, we selected those bacteria whose complete genome were available. In general, the genomes available in the databases correspond to the best-studied bacteria (e.g. Acidithiobacillus), therefore they were included in our analysis.

d. The content is plentiful, but some part of the reference literatures is kind of obsolete (in 5 years). Key publications should be cited as completed as possible.

Answer. In order to accomplish the requirement, the complete set of references was revised and checked. Finally, we managed to replace 15 references by equivalent and more recent works.

 

Metabolic characteristics of the bacteria used in this study (this table is included for purposes of revision only; in the manuscript we have just modified the S1 Table).

 Bacteria Carbon metabolism Optimal pH range Doi

1 Acidithiobacillus albertensis DSM14366 Autotrophic Acidophilic 10.1186/s40793-017-0282-y

2 Acidithiobacillus caldus ATCC51756 Autotrophic Acidophilic 10.1128/JB.00843-09

3 Acidithiobacillus ferridurans JCM18981 Autotrophic Acidophilic 10.1128/MRA.01028-18

4 Acidithiobacillus ferrivorans SS3 Autotrophic Acidophilic 10.1007/s00792-016-0882-2

5 Acidithiobacillus ferrooxidans ATCC23270 Autotrophic Acidophilic 10.3389/fmicb.2011.00079

6 Acidithiobacillus thiooxidans ATCC19377 Autotrophic Acidophilic 10.1186/s40793-017-0305-8

7 Acidithiobacillus sulfuriphilus CJ-2 Autotrophic Acidophilic 10.1099/ijsem.0.003576

8 Acidithiobacillus sp. PG05 Autotrophic Acidophilic 10.3389/fmicb.2022.960324

9 Acidithiobacillus ferrooxidans MC 2.2 Autotrophic Acidophilic 10.3389/fmicb.2022.960324

10 Acidiphilium angustum ATCC 35903 Heterotrophic Acidophilic 10.1099/00207713-36-2-197

11 Acidiphilium cryptum JF-5 Heterotrophic Acidophilic 10.1016/j.hydromet.2015.08.003

12 Acidiphilium multivorum AIU301 Heterotrophic Acidophilic 10.3390/microorganisms9050984

13 Acidiphilium rubrum ATCC 35905 Heterotrophic Acidophilic 10.1099/00207713-36-2-197

14 Ferrovum myxofaciens P3G Autotrophic Acidophilic 10.1128/AEM.03230-13

15 Ferrovum sp. JA12 Autotrophic Acidophilic 10.1371/journal.pone.0146832

16 Gallionella capsiferriformans ES-2 Autotrophic Neutrophilic 10.1111/gbi.12162

17 Sideroxydans lithotrophicus ES-1 Autotrophic Neutrophilic 10.1093/femsec/fiz034

18 Acidiferrobacter thiooxydans ZJ Autotrophic Acidophilic 10.1016/j.resmic.2018.08.001

19 Acidihalobacter ferrooxydans V8 Autotrophic Acidophilic 10.1128/genomeA.00413-17

20 Acidihalobacter prosperus DSM5130 Autotrophic Acidophilic 10.1128/genomeA.01042-14

21 Metallibacterium scheffleri DKE6 Mixotrophic Neutrophilic, acid-tolerant 10.1093/femsec/fix011

22 Escherichia coli O157:H7 str. Sakai DNA Heterotrophic Neutrophilic 10.1128/JB.01481-08

23 Escherichia coli BW25113 Heterotrophic Neutrophilic 10.1099/mic.0.000742

24 Mariprofundus ferroxydans O-1 Autotrophic Neutrophilic 10.1007/s12275-014-3625-z

25 Sulfurimonas autotrophica DSM16294 Autotrophic Neutrophilic 10.1099/ijs.0.02682-0

26 Sulfurimonas denitrificans DSM1251 Autotrophic Acidophilic 10.3389/fmicb.2015.00989

27 Leptospirillum ferrooxidans C2-3 Autotrophic Acidophilic 10.1128/jb.00696-12

28 Leptospirillum ferriphilum DSM14647 Autotrophic Acidophilic 10.1128/genomeA.01153-14

29 Leptospirillum sp. CF-1 Autotrophic Acidophilic 10.1016/j.jbiotec.2016.02.008

30 Leptospirillum rubarum Autotrophic Acidophilic 10.1128/AEM.02943-08

31 Acidibacillus sulfuroxidans Y002 Heterotrophic Acidophilic 10.1016/j.resmic.2022.104008

32 Acidibacillus ferrooxidans Huett2 Heterotrophic Acidophilic 10.4028/www.scientific.net/SSP.262.334

33 Alicyclobacillus acidocaldarius subsp. acidocaldarius Tc-4-1 Heterotrophic Acidophilic 10.1128/JB.05709-11

34 Alicyclobacillus tengchongensis CGMCC1504 Heterotrophic Acidophilic 10.1007/s12275-014-3625-z

35 Sulfobacillus acidophilus TPY Mixotrophic Acidophilic 10.3389/fmicb.2016.01861

36 Sulfobacillus thermosulfidooxidans DSM9293 Mixotrophic Acidophilic 10.3389/fmicb.2020.00044

37 Sulfobacillus thermotolerans Kr1 Mixotrophic Acidophilic 10.1038/s41598-019-51486-1

38 Sulfobacillus benefaciens Mixotrophic Acidophilic 10.1007/s00792-008-0184-4

39 Acidobacterium capsulatum ATCC 51196 Mixotrophic Acidophilic 10.1128/AEM.02294-08

40 Ferrimicrobium acidiphilum DSM 19497 Heterotrophic Acidophilic 10.1099/ijs.0.65409-0

41 Ferrithrix thermotolerans DSM 19514 Heterotrophic Acidophilic 10.1099/ijs.0.65409-0

42 Acidimicrobium ferrooxidans DSM 10331 Mixotrophic Acidophilic 10.4056/sigs.1463

---

## [Editor Report · Decision Letter 1]

24 Aug 2023

Comparative genomics of the proteostasis network in extreme acidophiles

PONE-D-23-16802R1

Dear Dr. Levicán,

We’re pleased to inform you that your manuscript has been judged scientifically suitable for publication and will be formally accepted for publication once it meets all outstanding technical requirements.

Kind regards,

RAJA AADIL HUSSAIN BHAT

Academic Editor

PLOS ONE
---

## [Editor Report · Acceptance letter]

29 Aug 2023

PONE-D-23-16802R1 

Comparative genomics of the proteostasis network in extreme acidophiles 

Dear Dr. Levicán:

I'm pleased to inform you that your manuscript has been deemed suitable for publication in PLOS ONE. Congratulations! Your manuscript is now with our production department. 

Kind regards, 

on behalf of

Dr. RAJA AADIL HUSSAIN BHAT 

Academic Editor

PLOS ONE